# Learning to Recombine and Resample Data for Compositional Generalization

**Ekin Akyürek**
MIT CSAIL
akyurek@mit.edu

**Afra Feyza Akyürek**
Boston University
akyurek@bu.edu

**Jacob Andreas**
MIT CSAIL
jda@mit.edu

## Abstract

Flexible neural sequence models outperform grammar- and automaton-based counterparts on a variety of tasks. However, neural models perform poorly in settings requiring compositional generalization beyond the training data—particularly to rare or unseen subsequences. Past work has found symbolic scaffolding (e.g. grammars or automata) essential in these settings. We describe R&R, a *learned data augmentation* scheme that enables a large category of compositional generalizations without appeal to latent symbolic structure. R&R has two components: *recombination* of original training examples via a prototype-based generative model and *resampling* of generated examples to encourage extrapolation. Training an ordinary neural sequence model on a dataset augmented with recombined and resampled examples significantly improves generalization in two language processing problems—instruction following (SCAN) and morphological analysis (SIGMORPHON 2018)—where R&R enables learning of new constructions and tenses from as few as eight initial examples.

## 1 Introduction

How can we build machine learning models with the ability to learn new concepts in context from little data? Human language learners acquire new word meanings from a single exposure (Carey & Bartlett, 1978), and immediately incorporate words and their meanings productively and compositionally into larger linguistic and conceptual systems (Berko, 1958; Piantadosi & Aslin, 2016). Despite the remarkable success of neural network models on many learning problems in recent years—including one-shot learning of classifiers and policies (Santoro et al., 2016; Wang et al., 2016)—this kind of few-shot learning of composable concepts remains beyond the reach of standard neural models in both diagnostic and naturalistic settings (Lake & Baroni, 2018; Bahdanau et al., 2019a).

Consider the *few-shot morphology learning* problem shown in Fig. 1, in which a learner must predict various linguistic features (e.g. 3rd person, SinGular, PRESent tense) from word forms, with only a small number of examples of the PAST tense in the training set. Neural sequence-to-sequence models (e.g. Bahdanau et al., 2015) trained on this kind of imbalanced data fail to predict past-tense tags on held-out inputs of any kind (Section 5). Previous attempts to address this and related shortcomings in neural models have focused on explicitly encouraging rule-like behavior by e.g. modeling data with symbolic grammars (Jia & Liang, 2016; Xiao et al., 2016; Cai et al., 2017) or applying rule-based data augmentation (Andreas, 2020). These procedures involve highly task-specific models or

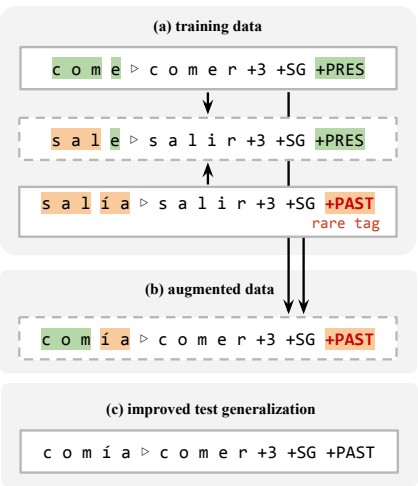

Figure 1: We first train a generative model to reconstruct training pairs $(x \triangleright y)$ by constructing them from other training pairs (a). We then perform data augmentation by *sampling* from this model, preferentially generating samples in which $y$ contains rare tokens or substructures (b). Dashed boxes show prediction targets. Conditional models trained on the augmented dataset accurately predict outputs $y$ from new inputs $x$ requiring compositional generalization (c).

generative assumptions, preventing them from generalizing effectively to less structured problems that combine rule-like and exceptional behavior. More fundamentally, they fail to answer the question of whether explicit rules are necessary for compositional inductive bias, and whether it is possible to obtain "rule-like" inductive bias *without* appeal to an underlying symbolic generative process.

This paper describes a procedure for improving few-shot compositional generalization in neural sequence models without symbolic scaffolding. Our key insight is that even fixed, imbalanced training datasets provide a rich source of supervision for few-shot learning of concepts and composition rules. In particular, we propose a new class of prototype-based neural sequence models (c.f. Gu et al., 2018) that can be directly trained to perform the kinds of generalization exhibited in Fig. 1 by explicitly **recombining** fragments of training examples to reconstruct other examples. Even when these prototype-based models are not effective as general-purpose predictors, we can **resample** their outputs to select high-quality synthetic examples of rare phenomena. Ordinary neural sequence models may then be trained on datasets augmented with these synthetic examples, distilling the learned regularities into more flexible predictors. This procedure, which we abbreviate **R&R**, promotes efficient generalization in both challenging synthetic sequence modeling tasks (Lake & Baroni, 2018) and morphological analysis in multiple natural languages (Cotterell et al., 2018).

By directly optimizing for the kinds of generalization that symbolic representations are supposed to support, we can bypass the need for symbolic representations themselves: R&R gives performance comparable to or better than state-of-the-art neuro-symbolic approaches on tests of compositional generalization. Our results suggest that some failures of systematicity in neural models can be explained by simpler structural constraints on data distributions and corrected with weaker inductive bias than previously described.[1]

## 2 BACKGROUND AND RELATED WORK

**Compositional generalization** Systematic compositionality—the capacity to identify rule-like regularities from limited data and generalize these rules to novel situations—is an essential feature of human reasoning (Fodor et al., 1988). While details vary, a common feature of existing attempts to formalize systematicity in sequence modeling problems (e.g. Gordon et al., 2020) is the intuition that learners should make accurate predictions in situations featuring novel combinations of previously observed input or output subsequences. For example, learners should generalize from actions seen in isolation to more complex commands involving those actions (Lake et al., 2019), and from relations of the form `r(a,b)` to `r(b,a)` (Keysers et al., 2020; Bahdanau et al., 2019b). In machine learning, previous studies have found that standard neural architectures fail to generalize systematically even when they achieve high in-distribution accuracy in a variety of settings (Lake & Baroni, 2018; Bastings et al., 2018; Johnson et al., 2017).

**Data augmentation and resampling** Learning to predict sequential outputs with rare or novel subsequences is related to the widely studied problem of *class imbalance* in classification problems. There, undersampling of the majority class or oversampling of the minority class has been found to improve the quality of predictions for rare phenomena (Japkowicz et al., 2000). This can be combined with targeted *data augmentation* with synthetic examples of the minority class (Chawla et al., 2002). Generically, given a training dataset $\mathcal{D}$, learning with class resampling and data augmentation involves defining an augmentation distribution $\tilde{p}(x, y \mid \mathcal{D})$ and sample weighting function $u(x, y)$ and maximizing a training objective of the form:

$$\mathcal{L}(\theta) = \underbrace{\frac{1}{|\mathcal{D}|} \sum_{x \in \mathcal{D}} \log p_\theta(y \mid x)}_{\text{Original training data}} + \underbrace{\mathbb{E}_{(x,y) \sim \tilde{p}} \, u(x, y) \log p_\theta(y \mid x)}_{\text{Augmented data}} \, . \tag{1}$$

In addition to task-specific model architectures (Andreas et al., 2016; Russin et al., 2019), recent years have seen a renewed interest in data augmentation as a flexible and model-agnostic tool for encouraging controlled generalization (Ratner et al., 2017). Existing proposals for sequence models are mainly rule-based—in sequence modeling problems, specifying a synchronous context-free grammar (Jia & Liang, 2016) or string rewriting system (Andreas, 2020) to generate new examples. Rule-based data augmentation schemes that recombine multiple training examples have been proposed

---

[1]Code for all experiments in this paper is available at `https://github.com/ekinakyurek/compgen`. We implemented our experiments in Knet (Yuret, 2016) using Julia (Bezanson et al., 2017).

for image classification (Inoue, 2018) and machine translation (Fadaee et al., 2017). While rule-based data augmentation is highly effective in structured problems featuring crisp correspondences between inputs and outputs, the effectiveness of such approaches involving more complicated, context-dependent relationships between inputs and outputs has not been well-studied.

**Learned data augmentation** What might compositional data augmentation look like without rules as a source of inductive bias? As Fig. 1 suggests, an ideal data augmentation procedure ($\tilde{p}$ in Eq. 1) should automatically identify valid ways of transforming and combining examples, without pre-committing to a fixed set of transformations.[2] A promising starting point is provided by **prototype-based** models, a number of which (Gu et al., 2018; Guu et al., 2018; Khandelwal et al., 2020) have been recently proposed for sequence modeling. Such models generate data according to:

$$d \sim p_{\text{rewrite}}(\cdot \mid d'; \theta) \quad \text{where} \quad d' \sim \text{Unif}(\mathcal{D}); \tag{2}$$

for a dataset $\mathcal{D}$ and a learned sequence rewriting model $p_{\text{rewrite}}(d \mid d'; \theta)$. (To avoid confusion, we will use the symbol $d$ to denote a *datum*. Because a data augmentation procedure must produce complete input–output examples, each $d$ is an $(x, y)$ pair for the conditional tasks evaluated in this paper.) While recent variants implement $p_{\text{rewrite}}$ with neural networks, these models are closely related to classical kernel density estimators (Rosenblatt, 1956). But additionally—building on the motivation in Section 1—they may be viewed as *one-shot learners* trained to generate new data $d$ from a single example.

Existing work uses prototype-based models as replacements for standard sequence models. We will show here that they are even better suited to use as data augmentation procedures: they can produce high-precision examples in the neighborhood of existing training data, then be used to bootstrap simpler predictors that extrapolate more effectively. But our experiments will also show that existing prototype-based models give mixed results on challenging generalizations of the kind depicted in Fig. 1 when used for either direct prediction or data augmentation—performing well in some settings but barely above baseline in others.

Accordingly, R&R is built on two model components that transform prototype-based language models into an effective learned data augmentation scheme. Section 3 describes an implementation of $p_{\text{rewrite}}$ that encourages greater sample diversity and well-formedness via a multi-prototype copying mechanism (a *two-shot* learner). Section 4 describes heuristics for sampling prototypes $d'$ and model outputs $d$ to focus data augmentation on the most informative examples. Section 5 investigates the empirical performance of both components of the approach, finding that they together provide they a simple but surprisingly effective tool for enabling compositional generalization.

## 3   PROTOTYPE-BASED SEQUENCE MODELS FOR DATA RECOMBINATION

We begin with a brief review of existing prototype-based sequence models. Our presentation mostly follows the *retrieve-and-edit* approach of Guu et al. (2018), but versions of the approach in this paper could also be built on retrieval-based models implemented with memory networks (Miller et al., 2016; Gu et al., 2018) or transformers (Khandelwal et al., 2020; Guu et al., 2020). The generative process described in Eq. 2 implies a marginal sequence probability:

$$p(d) = \frac{1}{|\mathcal{D}|} \sum_{d' \in \mathcal{D}} p_{\text{rewrite}}(d \mid d'; \theta) \tag{3}$$

Maximizing this quantity over the training set with respect to $\theta$ will encourage $p_{\text{rewrite}}$ to act as a model of *valid data transformations*: To be assigned high probability, every training example must be explained by at least one other example and a parametric rewriting operation. (The trivial solution where $p_\theta$ is the identity function, with $p_\theta(d \mid d' = d) = 1$, can be ruled out manually in

---

[2]As a concrete example of the potential advantage of learned data augmentation, consider applying the GECA procedure of Andreas (2020) to the language of strings $a^n b^n$. GECA produces a training set that is substitutable in the sense of Clark & Eyraud (2007); as noted there, $a^n b^n$ is not substitutable. GECA will infer that $a$ can be replaced with $aab$ based on their common context in ($a\underline{a}bb$, $a\underline{aa}bbb$), then generate the malformed example $a\underline{aa}babbb$ by replacing an $a$ in the wrong position. In contrast, recurrent neural networks can accurately model $a^n b^n$ (Weiss et al., 2018; Gers & Schmidhuber, 2001). Of course, this language can also be generated using even more constrained procedures than GECA, but in general learned sequence models can capture a broader set of both formal regularities and exceptions compared to rule-based procedures.

the design of $p_\theta$.) When $\mathcal{D}$ is large, the sum in Eq. 3 is too large to enumerate exhaustively when computing the marginal likelihood. Instead, we can optimize a lower bound by restricting the sum to a **neighborhood** $\mathcal{N}(d) \subset \mathcal{D}$ of training examples around each $d$:

$$p(d) \geq \frac{1}{|\mathcal{D}|} \sum_{d' \in \mathcal{N}(d)} p_{\text{rewrite}}(d \mid d'; \theta) . \tag{4}$$

The choice of $\mathcal{N}$ is discussed in more detail in Section 4. Now observe that:

$$\log p(d) \geq \log \left( |\mathcal{N}(d)| \sum_{d' \in \mathcal{N}(d)} \frac{1}{|\mathcal{N}(d)|} p_{\text{rewrite}}(d \mid d'; \theta) \right) - \log |\mathcal{D}| \tag{5}$$

$$\geq \frac{1}{|\mathcal{N}(d)|} \sum_{d' \in \mathcal{N}(d)} \log p_{\text{rewrite}}(d \mid d'; \theta) + \log \left( \frac{|\mathcal{N}(d)|}{|\mathcal{D}|} \right) \tag{6}$$

where the second step uses Jensen's inequality. If all $|\mathcal{N}(d)|$ are the same size, maximizing this lower bound on log-likelihood is equivalent to simply maximizing

$$\sum_{d' \in \mathcal{N}(d)} \log p_{\text{rewrite}}(d \mid d'; \theta) \tag{7}$$

over $\mathcal{D}$—this is the ordinary conditional likelihood for a string transducer (Ristad & Yianilos, 1998) or sequence-to-sequence model (Sutskever et al., 2014) with examples $d, d' \in \mathcal{N}(d)$.[3]

We have motivated prototype-based models by arguing that $p_{\text{rewrite}}$ learns a model of transformations licensed by the training data. However, when generalization involves complex compositions, we will show that neither a basic RNN implementation of $p_{\text{rewrite}}$ or a single prototype is enough; we must provide the learned rewriting model with a larger inventory of parts and encourage reuse of those parts as faithfully as possible. This motivates the two improvements on the prototype-based modeling framework described in the remainder of this section: generalization to multiple prototypes (Section 3.1) and a new rewriting model (Section 3.2).

### 3.1 $n$-PROTOTYPE MODELS

To improve *compositionality* in prototype-based models, we equip them with the ability to condition on multiple examples simultaneously. We extend the basic prototype-based language model to $n$ prototypes, which we now refer to as a **recombination** model $p_{\text{recomb}}$:

$$d \sim p_{\text{recomb}}(\cdot \mid d'_{1:n}; \theta) \quad \text{where} \quad d'_{1:n} \stackrel{\text{def}}{=} (d'_1, d'_2, \dots, d'_n) \sim p_\Omega(\cdot) \tag{9}$$

A multi-protype model may be viewed as a meta-learner (Thrun & Pratt, 1998; Santoro et al., 2016): it maps from a small number of examples (the prototypes) to a distribution over new datapoints consistent with those examples. By choosing the neighborhood and implementation of $p_{\text{recomb}}$ appropriately, we can train this meta-learner to specialize in one-shot concept learning (by reusing a fragment exhibited in a single prototype) or compositional generalization (by assembling fragments of prototypes into a novel configuration). To enable this behavior, we define a set of *compatible prototypes* $\Omega \subset \mathcal{D}^n$ (Section 4) and let $p_\Omega \stackrel{\text{def}}{=} \text{Unif}(\Omega)$. We update Eq. 6 to feature a corresponding multi-prototype neighborhood $\mathcal{N} : \mathcal{D} \to \Omega$. The only terms that have changed are the conditioning variable and the constant term, and it is again sufficient to choose $\theta$ to optimize $\sum_{d'_{1:n} \in \mathcal{N}(d)} \log p_{\text{recomb}}(d \mid d'_{1:n})$ over $\mathcal{D}$, implementing $p_{\text{recomb}}$ as described next.

### 3.2 RECOMBINATION NETWORKS

Past work has found that latent-variable neural sequence models often ignore the latent variable and attempt to directly model sequence marginals (Bowman et al., 2016). When an ordinary sequence-to-sequence model with attention is used to implement $p_{\text{recomb}}$, even in the one-prototype case, generated

---

[3]Past work also includes a continuous latent variable $z$, defining:

$$p_{\text{rewrite}}(d \mid d') = \mathbb{E}_{z \sim p(z)}[p_{\text{rewrite}}(d \mid d', z; \theta)] \tag{8}$$

As discussed in Section 5, the use of a continuous latent variable appears to make no difference in prediction performance for the tasks in this paper. The remainder of our presentation focuses on the simpler model in Eq. 7.

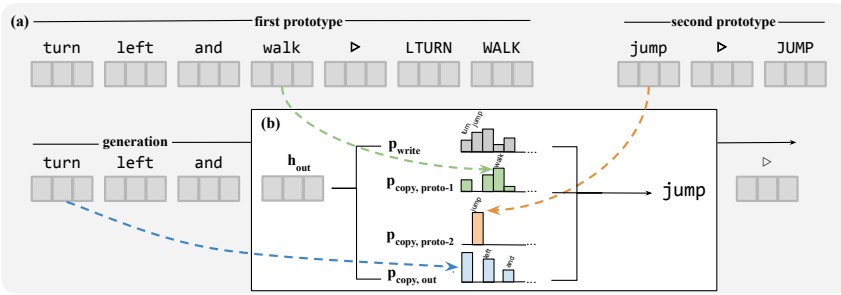

Figure 2: (a) RNN encoders produce contextual embeddings for prototype tokens. (b) In the decoder, a gated copy mechanism reuses prototypes and generated output tokens via an attention mechanism (dashed lines).

sentences often have little overlap with their prototypes (Weston et al., 2018). We describe a specific model architecture for $p_{\text{recomb}}$ that does not function as a generic noise model, and in which outputs are primarily generated via explicit reuse of fragments of multiple prototypes, by facilitating *copying* from independent streams containing prototypes and previously generated input tokens.

We take $p_{\text{recomb}}(d \mid d'_{1:n}; \theta)$ to be a neural (multi-)sequence-to-sequence model (c.f. Sutskever et al., 2014) which decomposes probability autoregressively: $p_{\text{recomb}}(d \mid d'_{1:n}; \theta) = \prod_t p(d^t \mid d^{<t}, d'_{1:n}; \theta)$. As shown in Fig. 2, three LSTM encoders—two for the prototypes and one for the input prefix—compute sequences of token representations $h_{\text{proto}}$ and $h_{\text{out}}$ respectively. Given the current decoder hidden state $h_{out}^t$, the model first attends to both prototype and output tokens:

$$\alpha_{\text{out}}^i \propto \exp(h_{\text{out}}^{t^\top} W_o\, h_{\text{out}}^i) \qquad i < t \tag{10}$$

$$\alpha_{\text{proto}}^{kj} \propto \exp(h_{\text{out}}^{t^\top} W_p\, h_{\text{proto}}^{kj}) \quad k \le n, j \le |d'_k| \tag{11}$$

To enable copying from each sequence, we project attention weights $\alpha_{\text{out}}$ and $\alpha_{\text{proto}}^k$ onto the output vocabulary to produce a sparse vector of probabilities:

$$p_{\text{copy,out}}^t(d^t = w) = \sum_{i<t} \mathbb{1}[d_i = w] \cdot \alpha_{\text{out}}^i \tag{12}$$

$$p_{\text{copy,proto-k}}^t(d^t = w) = \sum_{j \le |d'_k|} \mathbb{1}[d'_{k,j} = w] \cdot \alpha_{\text{proto}}^{kj} \tag{13}$$

Unlike rule-based data recombination procedures, however, $p_{\text{recomb}}$ is not required to copy from the prototypes, and can predict output tokens directly using *values* retrieved by the attention mechanism:

$$h_{\text{pre}}^t = \big[ h_{\text{out}}^t, \quad \sum_i \alpha_{\text{out}}^i h_{\text{out}}^i, \quad \sum_{k,j} \alpha_{\text{proto}}^{kj} h_{\text{proto}}^{kj} \big] \tag{14}$$

$$p_{\text{write}}^t \propto \exp(W_{\text{write}} h_{\text{pre}}^t) \tag{15}$$

To produce a final distribution over output tokens at time $t$, we combine predictions from each stream:

$$\beta_{\text{gate}} = \text{softmax}(W_{gate} h_{out}^t) \tag{16}$$

$$p(d^t = w \mid d^{<t}, d'_{1:n}; \theta) = \beta_{\text{gate}} \cdot [p_{\text{write}}^t(w), p_{\text{copy,out}}^t(w), p_{\text{copy,proto-1}}^t(w), ..., p_{\text{copy,proto-n}}^t(w)] \tag{17}$$

This copy mechanism is similar to the one proposed by Merity et al. (2017) and See et al. (2017). We compare 1- and 2-prototype models to an ordinary sequence model and baselines in Section 5.

## 4 SAMPLING SCHEMES

The models above provide generic procedures for generating well-formed combinations of training data, but do nothing to ensure that the generated samples are of a kind useful for compositional generalization. While the training objective in Eq. 7 encourages the learned $p(d)$ to lie close to the training data, an effective data augmentation procedure should intuitively provide *novel* examples of *rare* phenomena. To generate augmented training data, we combine the generative models of Section 3 with a simple sampling procedure that upweights useful examples.

### 4.1 RESAMPLING AUGMENTED DATA

In classification problems with imbalanced classes, a common strategy for improving accuracy on the rare class is to *resample* so that the rare class is better represented in training data (Japkowicz

et al., 2000). When constructing an augmented dataset using the models described above, we apply a simple rejection sampling scheme. In Eq. 1, we set:

$$u(d) = \mathbb{1}[\min_t p(d^t) < \epsilon] . \tag{18}$$

Here $p(d^t)$ is the marginal probability that the token $d^t$ appears in any example and $\epsilon$ is a hyperparameter. The final model is then trained using Eq. 1, retaining those augmented samples for which $u(d) = 1$. For extremely imbalanced problems, like the ones considered in Section 5, this weighting scheme effectively functions as a **rare tag constraint**: only examples containing rare words or tags are used to augment the original training data.

### 4.2 NEIGHBORHOODS AND PROTOTYPE PRIORS

How can we ensure that the data augmentation procedure generates any samples with positive weight in Eq. 18? The prototype-based models described in Section 3 offer an additional means of control over the generated data. Aside from the implementation of $p_{\text{recomb}}$, the main factors governing the behavior of the model are the choice of neighborhood function $\mathcal{N}(d)$ and, for $n \geq 2$, the set of prior compatible prototypes $\Omega$. Defining these so that rare tags also preferentially appear in prototypes helps ensure that the generated samples contribute to generalization. Let $d_1$ and $d_2$ be prototypes. As a notational convenience, given two sequences $d_1, d_2$, let $d_1 \backslash d_2$ the *set* of tokens in $d_1$ but not $d_2$, and $d_1 \Delta d_2$ denote the set of tokens not common to $d_1$ and $d_2$.

**1-prototype neighborhoods** Guu et al. (2018) define a one-prototype $\mathcal{N}$ based on a Jaccard distance threshold (Jaccard, 1901). For experiments with one-prototype models we employ a similar strategy, choosing an initial neighborhood of candidates such that

$$\mathcal{N}(d) \stackrel{\text{def}}{=} \{d_1 \in \mathcal{D} : (\alpha \cdot |d \Delta d_1| + \beta \cdot \text{lev}(d, d_1)) < \delta\} \tag{19}$$

where lev is string edit distance (Levenshtein, 1966) and $\alpha$, $\beta$ and $\delta$ are hyperparameters (discussed in Appendix B).

**2-prototype neighborhoods** The $n \geq 2$ prototype case requires a more complex neighborhood function—intuitively, for an input $d$, we want each $(d_1, d_2, \ldots)$ in the neighborhood to collectively contain enough information to reconstruct $d$. Future work might treat the neighborhood function itself as latent, allowing the model to identify groups of prototypes that make $d$ probable; here, as in existing one-prototype models, we provide heuristic implementations for the $n = 2$ case.

*Long–short recombination*: For each $(d_1, d_2) \in \mathcal{N}(d)$, $d_1$ is chosen to be similar to $d$, and $d_2$ is chosen to be similar to the *difference* between $d$ and $d_1$. (The neighborhood is so named because one of the prototypes will generally have fewer tokens than the other one.)

$$\mathcal{N}(d) \stackrel{\text{def}}{=} \{(d_1, d_2) \in \Omega : \text{lev}(d, d_1) < \delta, \text{lev}([d \backslash d_1], d_2) < \delta, |d \backslash d_1| > 0, |d \backslash d_1 \backslash d_2| = 0\} \tag{20}$$

Here $[d \backslash d_1]$ is the sequence obtained by removing all tokens in $d_1$ from $d$. Recall that we have defined $p_\Omega(d_{1:n}) \stackrel{\text{def}}{=} \text{Unif}(\Omega)$ for a set $\Omega$ of "compatible" prototypes. For experiments using long–short combination, all prototypes are treated as compatible; that is, $\Omega = \mathcal{D} \times \mathcal{D}$.

*Long–long recombination*: $\mathcal{N}(d)$ contains pairs of prototypes that are individually similar to $d$ and collectively contain all the tokens needed to reconstruct $d$:

$$\mathcal{N}(d) \stackrel{\text{def}}{=} \{(d_1, d_2) \in \Omega : \text{lev}(d, d_1) < \delta, \text{lev}(d, d_2) < \delta, |d \Delta d_1| = 1, |d \backslash d_1 \backslash d_2| = 0\} \tag{21}$$

For experiments using long–long recombination, we take $\Omega = \{(d_1, d_2) \in \mathcal{D} \times \mathcal{D} : |d_1 \Delta d_2| = 1\}$.

## 5 DATASETS & EXPERIMENTS

We evaluate R&R on two tests of compositional generalization: the SCAN instruction following task (Lake & Baroni, 2018) and a few-shot morphology learning task derived from the SIGMORPHON 2018 dataset (Kirov et al., 2018; Cotterell et al., 2018). Our experiments are designed to explore the effectiveness of learned data recombination procedures in controlled and natural settings. Both tasks involve conditional sequence prediction: while preceding sections have discussed augmentation

Table 1: Results on the SCAN dataset. (a) Comparison of R&R with previous work. Connecting lines indicate that model components are inherited from the parent (e.g. the row labeled *recomb-2* also includes resampling). Data augmentation with *recomb-2* + resampling performs slightly worse than GECA on the *jump* and *around right* splits; data augmentation with *recomb-1* + resampling or an ordinary RNN does not generalize robustly to either split. All differences except between GECA and *recomb-2* + resampling in *jump* are significant (paired $t$-test, $p \ll 0.001$). (Dashes indicate that all samples were rejected by resampling when decoding with temperature $T = 1$.) (b) Ablation experiments on the *jump* split. Introducing the latent variable used in previous work (Guu et al., 2018) does not change performance; removing the copy mechanism results in a complete failure of generalization. While it is possible to perform conditional inference of $p(y \mid x)$ given the generative model in Eq. 3 (*direct inference*), this gives significantly worse results than data augmentation (see Sec. 5.3).

(a)

| | *around right* | *jump* |
|---|---|---|
| baseline | 0.00 ±0.00 | 0.00 ±0.00 |
| GECA (published) | 0.82 ±0.11 | 0.87 ±0.05 |
| GECA (ours) | **0.98** ±0.02 | **1.00** ±0.001 |
| learned aug. (basic) | 0.00 ±0.00 | 0.00 ±0.00 |
| └ resampling | - | - |
| ├ *recomb-1* | 0.17 ±0.07 | - |
| └ *recomb-2* | 0.75 ±0.14 | 0.87 ±0.08 |
| *recomb-2* (no resampling) | 0.82 ±0.08 | 0.88 ±0.07 |

(b)

| | *jump* |
|---|---|
| *recomb-2* (no resampling) | **0.88** ±0.07 |
| + VAE | **0.88** ±0.07 |
| + resampling | **0.87** ±0.08 |
| − copying | 0.00 ±0.00 |
| direct inference | 0.57 ±0.05 |

procedures that produce data points $d = (x, y)$, learners are evaluated on their ability to predict an output $y$ from an input $x$: actions $y$ given instructions $x$, or morphological analyses $y$ given words $x$.

For each task, we compare a baseline with no data augmentation, the rule-based GECA data augmentation procedure (Andreas, 2020), and a sequence of ablated versions of R&R that measure the importance of resampling and recombination. The basic **Learned Aug** model trains an RNN to generate $(x, y)$ pairs, then trains a conditional model on the original data and samples from the generative model. **Resampling** filters these samples as described in Section 4. **Recomb-n** models replace the RNN with a prototype-based model as described in Section 3. Additional experiments (Table 1b) compare data augmentation to prediction of $y$ via **direct inference** (Appendix E) in the prototype-based model and several other model variants.

## 5.1 SCAN

SCAN (Lake & Baroni, 2018) is a synthetic dataset featuring simple English commands paired with sequences of actions. Our experiments aim to show that R&R performs well at one-shot concept learning and zero-shot generalization on controlled tasks where rule-based models succeed. We experiment with two splits of the dataset, *jump* and *around right*. In the *jump* split, which tests one-shot learning, the word *jump* appears in a single command in the training set but in more complex commands in the test set (e.g. *look and jump twice*). The *around right* split (Loula et al., 2018) tests zero-shot generalization by presenting learners with constructions like *walk around left* and *walk right* in the training set, but *walk around right* only in the test set.

Despite the apparent simplicity of the task, ordinary neural sequence-to-sequence models completely fail to make correct predictions on SCAN test set (Table 1). As such it has been a major focus of research on compositional generalization in sequence-to-sequence models, and a number of heuristic procedures and specialized model architectures and training procedures have been developed to solve it (Russin et al., 2019; Gordon et al., 2020; Lake, 2019; Andreas, 2020). Here we show that the generic prototype recombination procedure described above does so as well. We use long–short recombination for the *jump* split and long–long recombination for the *around right split*. We use a recombination network to generate 400 samples $d = (x, y)$ and then train an ordinary LSTM with attention (Bahdanau et al., 2019b) on the original and augmented data to predict $y$ from $x$. Training hyperparameters are provided in Appendix D.

Table 1 shows the results of training these models on the SCAN dataset.[4] 2-prototype recombination is essential for successful generalization on both splits. Additional ablations (Table 1b) show that the

---

[4]We provide results from GECA for comparison. Our final RNN predictor is more accurate than the one used by Andreas (2020), and training it on the same augmented dataset gives higher accuracies than reported in the original paper.

Table 2: $F_1$ score for morphological analysis on rare (FUT+PST) and frequent (OTHER) word forms. R&R variants with 1- and 2-prototype recombination (shaded in grey) consistently match or outperform both a no-augmentation baseline and GECA; *recomb-1* + resampling is best overall. Bold numbers are not significantly different from the best result in each column under a paired t-test ($p < 0.05$ after Bonferroni correction; nothing is bold if all differences are insignificant). The NOVEL portion of the table shows model accuracy on examples whose exact tag set never appeared in the training data. (There were no such words in the test set for the Spanish OTHER.) Differences between GECA and the best R&R variant (*recomb-1* + resampling) are larger than in the full evaluation set. *The Spanish past tense was used as a development set.

|  |  | Spanish | | Swahili | | Turkish | |
|---|---|---|---|---|---|---|---|
|  |  | FUT+PST* | OTHER | FUT+PST | OTHER | FUT+PST | OTHER |
| ALL | baseline | 0.66 ±0.01 | 0.88 ±0.01 | 0.75 ±0.02 | 0.90 ±0.01 | 0.69 ±0.04 | 0.85 ±0.03 |
|  | └ resampling | 0.65 ±0.01 | 0.88 ±0.01 | 0.77 ±0.01 | 0.90 ±0.02 | 0.69 ±0.04 | 0.84 ±0.04 |
|  | GECA | 0.66 ±0.01 | 0.88 ±0.01 | 0.76 ±0.02 | 0.90 ±0.02 | 0.69 ±0.02 | 0.87 ±0.01 |
|  | └ resampling | 0.72 ±0.02 | 0.88 ±0.01 | 0.81 ±0.02 | 0.89 ±0.01 | 0.75 ±0.03 | 0.85 ±0.02 |
|  | learned aug. (basic) | 0.66 ±0.02 | 0.88 ±0.01 | 0.77 ±0.02 | 0.90 ±0.01 | 0.70 ±0.02 | 0.87 ±0.01 |
|  | └ resampling | 0.70 ±0.02 | 0.86 ±0.01 | **0.84** ±0.02 | 0.90 ±0.01 | 0.73 ±0.02 | 0.85 ±0.03 |
|  | ├ *recomb-1* | 0.72 ±0.02 | 0.87 ±0.01 | **0.85** ±0.01 | 0.90 ±0.02 | 0.77 ±0.02 | 0.87 ±0.02 |
|  | └ *recomb-2* | 0.71 ±0.01 | 0.87 ±0.02 | 0.82 ±0.02 | 0.90 ±0.02 | 0.75 ±0.03 | 0.86 ±0.03 |
|  | GECA + *recomb-1* + resamp. | **0.74** ±0.02 | 0.86 ±0.01 | **0.85** ±0.02 | 0.89 ±0.01 | **0.79** ±0.02 | 0.84 ±0.01 |
| NOVEL | baseline | 0.63 ±0.03 | - | 0.72 ±0.02 | 0.42 ±0.12 | 0.68 ±0.04 | 0.66 ±0.15 |
|  | GECA + resampling | 0.67 ±0.03 | - | 0.79 ±0.02 | 0.26 ±0.20 | 0.73 ±0.04 | 0.71 ±0.10 |
|  | *recomb-1* + resampling | **0.69** ±0.02 | - | **0.83** ±0.02 | 0.42 ±0.12 | **0.75** ±0.03 | 0.82 ±0.04 |
|  | GECA + *recomb-1* + resamp. | **0.69** ±0.02 | - | **0.83** ±0.02 | 0.35 ±0.11 | **0.77** ±0.03 | 0.71 ±0.07 |

continuous latent variable used by Guu et al. (2018) does not affect performance, but that the copy mechanism described in Section 3.2 and the use of the *recomb-2* model for data augmentation rather than direct inference are necessary for accurate prediction.

## 5.2 SIGMORPHON 2018

The SIGMORPHON 2018 dataset consists of words paired with morphological analyses (*lemmas*, or base forms, and tags for linguistic features like tense and case, as depicted in Fig. 1). We use the data to construct a morphological *analysis* task (Akyürek et al., 2019) (predicting analyses from surface forms) to test models' few-shot learning of new morphological paradigms. In three languages of varying morphological complexity (Spanish, Swahili, and Turkish) we construct splits of the data featuring a training set of 1000 examples and three test sets of 100 examples. One test set consists exclusively of words in the past tense, one in the future tense and one with other word forms (present tense verbs, nouns and adjectives). The training set contains exactly eight past-tense and eight future-tense examples; all the rest are other word forms. Experiments evaluate R&R's ability to efficiently learn noisy morphological rules, long viewed a key challenge for connectionist approaches to language learning (Rumelhart & McClelland, 1986). As approaches may be sensitive to the choice of the eight examples from which the model must generalize, we construct five different splits per language and use the Spanish past-tense data as a development set. As above, we use *long–long* recombination with similarity criteria applied to $y$ only. We augment the training data with 180 samples from $p_{recomb}$ and again train an ordinary LSTM with attention for final predictions. Details are provided in Appendix B.

Table 2 shows aggregate results across languages. We report the model's $F_1$ score for predicting morphological analyses of words in the *few-shot* training condition (past and future) and the standard training condition (other word forms). Here, learned data augmentation with both one- and two-prototype models consistently matches or outperforms GECA. The improvement is sometimes dramatic: for few-shot prediction in Swahili, *recomb-1* augmentation reduces the error rate by 40% relative to the baseline and 21% relative to GECA. An additional **baseline + resampling** experiment upweights the existing rare samples rather than synthesizing new ones; results demonstrate that recombination, and not simply reweighting, is important for generalization. Table 2 also includes a finer-grained analysis of *novel word forms*: words in the evaluation set whose exact morphological analysis never appeared in the training set. R&R again significantly outperforms both the baseline and GECA-based data augmentation in the few-shot FUT+PAST condition and the ordinary OTHER condition, underscoring the effectiveness of this approach for "in-distribution" compositional

generalization. Finally, the gains provided by learned augmentation and GECA appear to be at least partially orthogonal: combining the GECA + resampling and *recomb-1* + resampling models gives further improvements in Spanish and Turkish.

## 5.3 ANALYSIS

**Why is R&R effective?** Samples from the best learned data augmentation models for SCAN and SIGMORPHON may be found in the Appendix G.3 . We programaticaly analyzed 400 samples from *recomb-2* models in SCAN and found that 40% of novel samples are exactly correct in the *around right* split and 74% in the *jump* split. A manual analysis of 50 Turkish samples indicated that only 14% of the novel samples were exactly correct. The augmentation procedure has a high error rate! However, our analysis found that malformed samples either (1) feature malformed $x$s that will never appear in a test set (a phenomenon also observed by Andreas (2020) for outputs of GECA), or (2) are mostly correct at the token level (inducing predictions with a high $F_1$ score). Data augmentation thus contributes a mixture of irrelevant examples, *label noise*—which may exert a positive regularizing effect (Bishop, 1995)—and well-formed examples, a small number of which are sufficient to induce generalization (Bastings et al., 2018). Without resampling, SIGMORPHON models generate almost no examples of rare tags.

**Why does R&R outperform direct inference?** A partial explanation is provided by the preceding analysis, which notes that the accuracy of the data augmentation procedure as a generative model is comparatively low. Additionally, the data augmentation procedure selects only the highest-confidence samples from the model, so the quality of predicted $y$s conditioned on random $x$s will in general be even lower. A conditional model trained on augmented data is able to compensate for errors in augmentation or direct inference (Table 12 in the Appendix).

**Why is Resampling without Recombination effective?** One surprising feature of Table 2 is performance of the *learned aug (basic)* + *resampling* model. While less effective than the recombination-based models, augmentation with samples from an ordinary RNN trained on $(x, y)$ pairs improves performance for some test splits. One possible explanation is that resampling effectively acts as a *posterior constraint* on the final model's predictive distribution, guiding it toward solutions in which rare tags are more probable than observed in the original training data. Future work might model this constraint explicitly, e.g. via posterior regularization (as in Li & Rush, 2020).

## 6 CONCLUSIONS

We have described a method for improving compositional generalization in sequence-to-sequence models via data augmentation with learned prototype recombination models. These are the first results we are aware of demonstrating that generative models of data are effective as data augmentation schemes in sequence-to-sequence learning problems, even when the generative models are themselves unreliable as base predictors. Our experiments demonstrate that it is possible to achieve compositional generalization on-par with complex symbolic models in clean, highly structured domains, and outperform them in natural ones, with basic neural modeling tools and without symbolic representations.

ACKNOWLEDGMENTS

We thank Eric Chu for feedback on early drafts of this paper. This work was supported by a hardware donation from NVIDIA under the NVAIL grant program. The authors acknowledge the MIT SuperCloud and Lincoln Laboratory Supercomputing Center (Reuther et al., 2018) for providing HPC resources that have contributed to the research results reported within this paper.

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

# A  MODEL ARCHITECTURE

## A.1  PROTOTYPE ENCODER

We use a single layer BiLSTM network to encode $h_{\text{proto}}^{kj}$ as follows:

$$h_{\text{proto}}^{k} = \text{proj}(\text{BiLSTM}\,(W_e\,d'_k)) \tag{22}$$

**Morphology**  The hidden and embedding sizes are 1024. No dropout is applied. We project bi-directional embeddings to the hidden size with a linear projection. We concatenate the backward and the forward hidden states.

**SCAN**  We choose the hidden size as 512, and embedding size as 64. We apply 0.5 dropout to the input. We project hidden vectors in the attention mechanism.

## A.2  DECODER

The decoder is implemented by a single layer. In addition to the hidden state and memory cell, we also carry out a *feed* vector through time:

$$h_{\text{pre}}^{t} = \left[h_{\text{out}}^{t}, \quad \sum_{i} \alpha_{\text{out}}^{i} h_{\text{out}}^{i}, \quad \sum_{j \leq |d_1|} \alpha_{\text{proto}}^{1j} h_{\text{proto}}^{1j}, \quad \cdots, \quad \sum_{j \leq |d_n|} \alpha_{\text{proto}}^{nj} h_{\text{proto}}^{nj} \right] \tag{23}$$

$$\text{feed}^{t} = \text{Linear}_{\text{feed}}(h_{\text{pre}}^{t}) \tag{24}$$

The input to the LSTM decoder at time step $t$ is the concatenation of the previous token's representation, previous feed vector, and a latent $z$ vector (in the VAE model).

$$\text{input}^{t} = [W_d d^{t-1}, feed^{t-1}, z] \tag{25}$$

**Morphology**  We use a single-layer LSTM network with a hidden size of 1024, and an embedding size of 1024. We initialize the decoder hidden states with the final hidden states of the BiLSTM encoder. *feed* is the same size as the hidden state. No dropout is applied in the decoder. Output calculations are provided in the original paper in equation Eq. 16. The query vector for the attentions is identically the hidden state:

$$\text{query}^{t} = h_{\text{out}}^{t} \tag{26}$$

Further details of the attention are provided in Appendix A.3.

**SCAN**  The decoder is implemented by a single layer LSTM network with hidden size of 512, and embedding size of 64. The embedding parameters are shared with the encoder. Here the size of the feed vector is equal to embedding size, 64.

We have no self-attention for this decoder in the feed vector. There is an attention projection with dimension is 128. The details of the attention mechanism given in Appendix A.3. Finally, we use transpose of the embedding matrix to project *feed* to the output space.

$$\text{output}^{t} = W_e^{\top} \text{feed}^{t} \tag{27}$$

$output^{t}$ contains unnormalized scores before the final softmax layer. We apply 0.7 dropout to $h_{out}^{t}$ during both training and test. The copy mechanism will be further described in Appendix A.3.

The input to the LSTM decoder is the same as Eq. 25 except the decoder embedding matrix, $W_d$, shares parameters with encoder embedding matrix $W_e$. We applied 0.5 dropout to the embeddings $d_{t-1}$.

The *query* vector for the attention is calculated by:

$$\text{query}^{t} = [h_{\text{out}}^{t}, \text{input}^{t}] \tag{28}$$

## A.3 ATTENTION AND COPYING

We use the attention mechanism described in Vaswani et al. (2017) with slight modifications.

**Morphology** We use a linear transformation for *key* while retaining an embedding size of 1024, and leave *query* and *value* transformations as the identity. We do not normalize by the square root of the attention dimension. The query vector is described in the decoder Appendix A.2. The copy mechanism for the morphology task is explained in the paper in detail.

**SCAN** We use the nonlinear `tanh` transformation for *key*, *query* and *value*. That the attention scores are calculated separately for each prototype using different parameters as well as the normalization i.e. obtaining $\alpha$'s is performed separately for each prototype.

The copy mechanism for this task is slightly different and follows Gu et al. (2016) We normalize prototype attention scores and output scores jointly. Let $\bar{\alpha}_i$ represent attention weights for each prototype sequence *before* normalization. Then, we concatenate them to the output vector in Eq. 27.

$$\text{final}^t = [\text{output}^t, \bar{\alpha}_1, ..., \bar{\alpha}_n]$$

We obtain a probability vector via a final softmax layer:

$$\text{prob}^t = \text{softmax}(final^t)$$

That size of this probability vector is vocabulary size plus the total length all prototypes. We then project this into the output space by:

$$p^t(w) = \text{prob}^t(\texttt{indices}(w))$$

where `indices` finds all corresponding scores in $prob^t$ for token $w$ where there might be more than one element for a given $w$. This is because one score can come from the $output^t$ region, and others from the prototype regions of $prob^t$. During training we applied 0.5 dropout to the indices from $output^t$. Thus, the model is encouraged to copy more.

## B NEIGHBORHOODS AND SAMPLING

In the Eq. 20 and Eq. 21 we expressed the generic form of neighborhood sets. Here we provide the implementation details.

**SCAN** In the *jump* split, we use long-short recombination with $\delta = 0.5$. In *around right* we use long-long recombination with $\delta = 0.5$, and construct $\Omega$ so that the first and second prototypes to differ by a single token. We randomly pick $k < 10 \times 3$ (10 different first prototypes, and 3 different second prototypes for each of them) prototype pairs that satisfy these conditions. For the *recomb-1* experiment, we use the same neighborhood setup except but consider only the $k < 10$ first prototypes.

**Sampling** In the *jump* split, we used beam search with beam size 4 in the decoder. We calculate the mean and standard deviation over the lengths of both among the first $d'_1$ and the second $d'_2$ prototypes in the train set. Then, during the sampling, we expect the first and second prototypes whose length is shorter than their respective mean plus standard deviation. This decision is based on the fact that the part of the $\Omega$ that the model is exposed to is determined by the empirical distribution, $\hat{\Omega}$, that arises from training neighborhoods. When sampling, we try to pick prototypes from a distribution that are close to properties of that empirical distribution. In *around right*, we use temperature sampling with $T = 0.4$. If a model cannot sample the expected number of both novel and unique samples within a reasonable time, we increase temperature $T$.

**Morphology** We use long-long recombination, as explained in the paper, with slight modifications which leverage the structure of the task. We set $\Omega$ as:

$$\Omega = \{(d_1, d_2) \,|\, d'_{1\text{tags}} \neq d'_{2tags}, (d_{\text{tags}} \backslash d'_{1\text{tags}} \backslash d'_{2\text{tags}}) = 0\}$$

For the *recomb-1* model $\mathcal{N}(d)$ utilizes tag similarity, lemma similarity and is constructed using a score function:

$$\text{score}_1(d, d'_1) = (|d_{\text{tags}} \Delta d'_{1\text{tags}}|, \text{jaccard}(d_{\text{lemma}}, d'_{1\text{lemma}})) \tag{29}$$

Given $d$, we sort training examples by using $score_1$ as the comparison key and pick the four smallest neighbors (using a lexicographic sort) to form $\mathcal{N}(d)$.

For the *recomb-2* model, $\mathcal{N}(d)$ uses the same score function for the first prototype as in the *recomb-1* case. The second prototype is selected using:

$$\text{score}_2(d, d'_1, d'_2) = \left(d_{\text{tags}} \neq d'_{2\text{tags}}, |d'_{1\text{tags}} \Delta d'_{2\text{tags}}|\right) \tag{30}$$

Given $x$, and a scored first prototype, we do one more sort over training examples by using $\text{score}_2$ as the comparison key. Then we pick first four neighbors for $\mathcal{N}(d)$.

**Sampling** We use a mix strategy of temperature sampling with $T = 0.5$ and greedy sampling in which we use the former for $d_{\text{input}}$ and the latter for $d_{\text{output}}$. We sample 180 unique and novel examples.

## C   GENERATIVE MODEL TRAINING

**Morphology** All of the hyper parameters mentioned here are optimized by a grid search on the Spanish validation set. We train our models for 25 epochs[5]. We use Adam optimizer with learning rate 0.0001. The generative model is trained on morphological reinflection order ($d_{\text{lemma}}d_{\text{tags}} \triangleright d_{\text{inflection}}$) from left to right, then the samples from the model are reordered for morphological analysis task ($d_{\text{inflection}} \triangleright d_{\text{lemma}}d_{\text{tags}}$).

**SCAN** We use different number of epochs for *jump* and *around right* splits where all models are trained for 8 epochs in the former and 3 epochs in the latter. We use Adam optimizer with learning rate 0.002, and gradient norm clip with 1.0.

## D   SEQ2SEQ BASELINE MODEL

After generating novel samples, we either concatenate them to the training data (in morphology), or sample training batches from a mixture of the original training data and the augmented data. Our conditional model is the same as the generative model used in morphology experiments, described in detail in the paper body, replacing $d_{\text{proto}}$ with $x$, and $d$ with $y$.

Every conditional model's size is the same as the corresponding generative model which was used for augmentation. This is to ensure that the conditional model and the generative model have the same capacity. We train conditional models for 150 epochs for SCAN and we used augmentation ratios of $p_{\text{aug}} = 0.01$ and $p_{\text{aug}} = 0.2$ in *jump* and *around right*, respectively. For morphology, we train the conditional models for 100 epochs, and we use all generated examples for augmentation.

## E   DIRECT INFERENCE

To adapt the prototype-based model for conditional prediction, we condition the neighborhood function on the input $x$ rather than the full datum $d$, as in Hashimoto et al. (2018). Candidate $y$s are then sampled from the generative model given the observed $x$ while marginalizing over retrieved prototypes. Finally, we re-rank these candidates via Eq. 7 and output the highest-scoring candidate.

## F   VAE MODEL

**Prior** $p(z)$: We use the same prior as Guu et al. (2018) given in Eq. 31. In this prior, $z$ is defined by a norm and direction vector. The norm is sampled from the uniform distribution between zero and a maximum possible norm $\mu_{\text{max}} = 10.0$, and the direction is sampled uniformly from the unit hypersphere. This sampling procedure corresponds to a von Mises–Fisher distribution with concentration parameter zero.

---

[5]When training 2-proto and 1-proto models, we increment epoch counter when the entire neighborhood for every $d$ is processed. For 0-proto, one epoch is defined canonically i.e. the entire train set.

$$z = z_{\text{norm}} \cdot z_{\text{dir}} \quad where \quad z_{\text{norm}} \sim U(0, \mu_{\text{max}}), \quad z_{\text{dir}} \sim \text{vmF}(\vec{u}, 0) \tag{31}$$

For SCAN, the size of $z$ is 32, and for morphology the size of $z$ is 2.

**Proposal Network** $q(z|d, d_{1:n})$**:** Similarly to the prior, the posterior network decomposes $z$ into its norm and direction vectors. The norm vector is sampled from a uniform distribution at $(|\mu|, \min(|\mu| + \epsilon, \mu_{\text{max}}))$, and the direction is sampled from the von Mises–Fisher distribution $\text{vmF}(\mu, \kappa)$ where $\kappa = 25, \epsilon = 1.0$.

$$h_{\text{final}} = (\overleftarrow{h}_{\text{proto}})_{\text{start}} + (\overrightarrow{h}_{\text{proto}})_{end}$$
$$\mu = \tanh\left(W_z \, [h_{d\text{final}}, h_{q\text{final}}]\right)$$
$$z_{\text{norm}} \sim U(|\mu|, min(|\mu| + \epsilon, \mu_{\text{max}})) \tag{32}$$
$$z_{\text{dir}} \sim vmF(\mu, \kappa) \tag{33}$$
$$z = z_{\text{norm}} \cdot z_{\text{dir}} \tag{34}$$

# G  ADDITIONAL RESULTS

## G.1  MORPHOLOGY RESULTS

In the paper, Table 2 shows morphology results for (non-VAE) models with 8 *hints* (past- and future-tense examples in the training set). Here, we provide additional results for different hint set sizes and model variants.

### G.1.1  HINTS=4

Table 3: Exact Match Accuracy

|  | Spanish | | Swahili | | Turkish | |
|---|---|---|---|---|---|---|
|  | FUT+PST* | OTHER | FUT+PST | OTHER | FUT+PST | OTHER |
| baseline | 0.078 ±0.029 | 0.63 ±0.09 | 0.107 ±0.034 | 0.532 ±0.029 | 0.067 ±0.020 | 0.57 ±0.04 |
| geca | 0.072 ±0.019 | 0.63 ±0.05 | 0.039 ±0.011 | 0.496 ±0.027 | 0.052 ±0.014 | 0.54 ±0.08 |
| geca + resampling | 0.16 ±0.04 | 0.65 ±0.05 | 0.27 ±0.08 | 0.52 ±0.04 | 0.12 ±0.04 | 0.554 ±0.029 |
| learned aug | 0.063 ±0.012 | 0.65 ±0.04 | 0.066 ±0.034 | 0.52 ±0.04 | 0.074 ±0.021 | 0.57 ±0.04 |
| learned aug + resampling | 0.098 ±0.021 | 0.65 ±0.05 | 0.29 ±0.06 | 0.480 ±0.035 | 0.092 ±0.029 | 0.54 ±0.06 |
| recomb-1 | 0.063 ±0.017 | 0.674 ±0.021 | 0.061 ±0.017 | 0.520 ±0.028 | 0.055 ±0.021 | 0.554 ±0.030 |
| recomb-1 + resampling | 0.13 ±0.04 | 0.64 ±0.04 | 0.29 ±0.04 | 0.48 ±0.04 | 0.15 ±0.04 | 0.52 ±0.06 |
| recomb-2 | 0.061 ±0.010 | 0.656 ±0.030 | 0.08 ±0.06 | 0.524 ±0.026 | 0.073 ±0.019 | 0.58 ±0.05 |
| recomb-2 + resampling | 0.108 ±0.021 | 0.64 ±0.05 | 0.18 ±0.04 | 0.542 ±0.035 | 0.067 ±0.026 | 0.55 ±0.06 |

Table 4: F1 Accuracy

|  | Spanish | | Swahili | | Turkish | |
|---|---|---|---|---|---|---|
|  | FUT+PST* | OTHER | FUT+PST | OTHER | FUT+PST | OTHER |
| baseline | 0.609 ±0.025 | 0.873 ±0.034 | 0.746 ±0.013 | 0.897 ±0.005 | 0.561 ±0.032 | 0.867 ±0.015 |
| geca | 0.606 ±0.019 | 0.871 ±0.017 | 0.722 ±0.018 | 0.884 ±0.007 | 0.565 ±0.034 | 0.856 ±0.028 |
| geca + resampling | 0.675 ±0.017 | 0.870 ±0.022 | 0.802 ±0.023 | 0.892 ±0.010 | 0.65 ±0.05 | 0.850 ±0.019 |
| learned aug | 0.597 ±0.021 | 0.871 ±0.024 | 0.737 ±0.010 | 0.897 ±0.011 | 0.58 ±0.04 | 0.853 ±0.026 |
| learned aug + resampling | 0.646 ±0.007 | 0.872 ±0.028 | 0.826 ±0.013 | 0.887 ±0.011 | 0.637 ±0.032 | 0.835 ±0.034 |
| recomb-1 | 0.596 ±0.020 | 0.884 ±0.010 | 0.727 ±0.012 | 0.893 ±0.010 | 0.557 ±0.032 | 0.868 ±0.010 |
| recomb-1 + resampling | 0.663 ±0.029 | 0.874 ±0.014 | 0.812 ±0.017 | 0.886 ±0.011 | 0.67 ±0.04 | 0.84 ±0.04 |
| recomb-2 | 0.598 ±0.019 | 0.874 ±0.007 | 0.730 ±0.023 | 0.894 ±0.008 | 0.581 ±0.035 | 0.865 ±0.024 |
| recomb-2 + resampling | 0.658 ±0.012 | 0.872 ±0.015 | 0.778 ±0.011 | 0.897 ±0.007 | 0.609 ±0.032 | 0.850 ±0.029 |

### G.1.2  HINTS=8

Main 8-prototype $F_1$ results are provided in the body of the paper. Here we provide exact match results and an extra set of comparisons to the VAE model.

Table 5: Exact Match Accuracy

|  | Spanish | | Swahili | | Turkish | |
|---|---|---|---|---|---|---|
|  | FUT+PST* | OTHER | FUT+PST | OTHER | FUT+PST | OTHER |
| baseline | 0.151 ±0.017 | 0.65 ±0.04 | 0.15 ±0.04 | 0.554 ±0.034 | 0.23 ±0.06 | 0.55 ±0.04 |
| geca | 0.136 ±0.030 | 0.638 ±0.026 | 0.15 ±0.05 | 0.55 ±0.06 | 0.21 ±0.05 | 0.550 ±0.032 |
| geca + resampling | 0.249 ±0.034 | 0.64 ±0.04 | 0.25 ±0.05 | 0.532 ±0.033 | 0.27 ±0.07 | 0.524 ±0.026 |
| learned aug | 0.163 ±0.030 | 0.652 ±0.033 | 0.18 ±0.05 | 0.560 ±0.026 | 0.23 ±0.04 | 0.548 ±0.019 |
| learned aug + resampling | 0.181 ±0.026 | 0.590 ±0.032 | 0.34 ±0.06 | 0.552 ±0.029 | 0.24 ±0.04 | 0.53 ±0.05 |
| recomb-1 | 0.155 ±0.018 | 0.628 ±0.020 | 0.161 ±0.017 | 0.560 ±0.025 | 0.22 ±0.04 | 0.538 ±0.025 |
| recomb-1 + resampling | 0.218 ±0.032 | 0.616 ±0.034 | 0.35 ±0.04 | 0.53 ±0.04 | 0.30 ±0.04 | 0.52 ±0.04 |
| recomb-2 | 0.131 ±0.028 | 0.634 ±0.027 | 0.19 ±0.11 | 0.56 ±0.04 | 0.24 ±0.05 | 0.528 ±0.032 |
| recomb-2 + resampling | 0.203 ±0.035 | 0.63 ±0.05 | 0.27 ±0.07 | 0.552 ±0.031 | 0.25 ±0.05 | 0.54 ±0.06 |

Table 6: $F_1$ Accuracy (VAE model)

|  | Spanish | | Swahili | | Turkish | |
|---|---|---|---|---|---|---|
|  | FUT+PST* | OTHER | FUT+PST | OTHER | FUT+PST | OTHER |
| learned aug + resampling +vae | 0.689 ±0.018 | 0.859 ±0.010 | 0.845 ±0.014 | 0.896 ±0.011 | 0.730 ±0.032 | 0.850 ±0.015 |
| recomb-1 + resampling +vae | 0.717 ±0.014 | 0.870 ±0.007 | 0.843 ±0.014 | 0.898 ±0.010 | 0.736 ±0.030 | 0.859 ±0.031 |
| recomb-2 + resampling +vae | 0.710 ±0.008 | 0.865 ±0.012 | 0.824 ±0.015 | 0.896 ±0.011 | 0.751 ±0.027 | 0.848 ±0.027 |

### G.1.3 HINTS=16

Table 7: Exact Match Accuracy

|  | Spanish | | Swahili | | Turkish | |
|---|---|---|---|---|---|---|
|  | FUT+PST* | OTHER | FUT+PST | OTHER | FUT+PST | OTHER |
| baseline | 0.27 ±0.05 | 0.65 ±0.04 | 0.28 ±0.06 | 0.544 ±0.029 | 0.40 ±0.04 | 0.614 ±0.032 |
| geca | 0.26 ±0.06 | 0.65 ±0.06 | 0.26 ±0.05 | 0.530 ±0.028 | 0.37 ±0.05 | 0.570 ±0.035 |
| geca + resampling | 0.34 ±0.05 | 0.63 ±0.04 | 0.32 ±0.07 | 0.506 ±0.034 | 0.42 ±0.05 | 0.590 ±0.035 |
| learned aug | 0.25 ±0.04 | 0.65 ±0.04 | 0.32 ±0.04 | 0.538 ±0.028 | 0.39 ±0.04 | 0.58 ±0.05 |
| learned aug + resampling | 0.230 ±0.035 | 0.61 ±0.04 | 0.42 ±0.06 | 0.54 ±0.04 | 0.42 ±0.05 | 0.578 ±0.027 |
| recomb-1 | 0.27 ±0.05 | 0.63 ±0.06 | 0.32 ±0.05 | 0.55 ±0.04 | 0.35 ±0.06 | 0.60 ±0.05 |
| recomb-1 + resampling | 0.28 ±0.04 | 0.61 ±0.07 | 0.418 ±0.035 | 0.548 ±0.023 | 0.35 ±0.06 | 0.56 ±0.04 |
| recomb-2 | 0.22 ±0.06 | 0.62 ±0.07 | 0.28 ±0.04 | 0.56 ±0.04 | 0.40 ±0.06 | 0.596 ±0.024 |
| recomb-2 + resampling | 0.262 ±0.025 | 0.61 ±0.07 | 0.405 ±0.028 | 0.53 ±0.04 | 0.43 ±0.06 | 0.61 ±0.04 |

Table 8: F1 Accuracy

|  | Spanish | | Swahili | | Turkish | |
|---|---|---|---|---|---|---|
|  | FUT+PST* | OTHER | FUT+PST | OTHER | FUT+PST | OTHER |
| baseline | 0.733 ±0.014 | 0.881 ±0.012 | 0.811 ±0.018 | 0.893 ±0.011 | 0.750 ±0.026 | 0.875 ±0.021 |
| geca | 0.736 ±0.019 | 0.884 ±0.018 | 0.800 ±0.024 | 0.889 ±0.012 | 0.74 ±0.04 | 0.863 ±0.019 |
| geca + resampling | 0.782 ±0.024 | 0.867 ±0.012 | 0.830 ±0.021 | 0.885 ±0.013 | 0.794 ±0.032 | 0.865 ±0.018 |
| learned aug | 0.738 ±0.020 | 0.877 ±0.008 | 0.816 ±0.024 | 0.893 ±0.012 | 0.752 ±0.024 | 0.868 ±0.020 |
| learned aug + resampling | 0.745 ±0.019 | 0.870 ±0.012 | 0.866 ±0.016 | 0.894 ±0.013 | 0.787 ±0.031 | 0.863 ±0.021 |
| recomb-1 | 0.738 ±0.021 | 0.877 ±0.019 | 0.820 ±0.018 | 0.896 ±0.014 | 0.735 ±0.033 | 0.874 ±0.026 |
| recomb-1 + resampling | 0.770 ±0.020 | 0.867 ±0.023 | 0.872 ±0.005 | 0.892 ±0.010 | 0.778 ±0.024 | 0.861 ±0.022 |
| recomb-2 | 0.716 ±0.019 | 0.876 ±0.022 | 0.815 ±0.017 | 0.897 ±0.016 | 0.752 ±0.034 | 0.873 ±0.017 |
| recomb-2 + resampling | 0.765 ±0.023 | 0.868 ±0.021 | 0.856 ±0.015 | 0.888 ±0.016 | 0.808 ±0.018 | 0.868 ±0.027 |

## G.2 SIGNIFICANCE TESTS

Tables 9, 10 and 11 sho the $p$-values for pairwise differences between the baseline and prototype-based models

Table 9: Turkish language $p$-values for paired $t$-test in PST+FUT tenses for the average $F_1$ (micro) scores over several runs without Bonferronni correction.

|  | baseline | geca | learned aug | recomb-1 | recomb-2 | geca + resampling | learned aug + resampling | recomb-1 + resampling | recomb-2 + resampling |
|---|---|---|---|---|---|---|---|---|---|
| baseline |  |  |  |  |  |  |  |  |  |
| geca | 0.259314 |  |  |  |  |  |  |  |  |
| learned aug | 0.352506 | 0.802058 |  |  |  |  |  |  |  |
| recomb-1 | 0.707534 | 0.129244 | 0.187597 |  |  |  |  |  |  |
| recomb-2 | 0.233578 | 0.0230554 | 0.0331794 | 0.363375 |  |  |  |  |  |
| geca + resampling | 1.0125e-16 | 3.7044e-12 | 4.07678e-15 | 6.04788e-17 | 1.71167e-19 |  |  |  |  |
| learned aug + resampling | 8.00807e-10 | 1.37553e-07 | 6.51548e-08 | 6.35314e-11 | 3.76501e-13 | 0.0167999 |  |  |  |
| recomb-1 + resampling | 3.85877e-26 | 1.60117e-20 | 2.76421e-22 | 6.41228e-26 | 2.07776e-26 | 0.0109365 | 1.948e-06 |  |  |
| recomb-2 + resampling | 2.56689e-15 | 3.4083e-13 | 2.08177e-14 | 2.92113e-18 | 1.79928e-19 | 0.981886 | 0.0190462 | 0.0101878 |  |

Table 10: Spanish language $p$-values for paired $t$-test in PST+FUT tenses for the average $F_1$ (micro) scores over several runs without Bonferronni correction.

| | baseline | geca | learned aug | recomb-1 | recomb-2 | geca + resampling | learned aug + resampling | recomb-1 + resampling | recomb-2 + resampling |
|---|---|---|---|---|---|---|---|---|---|
| baseline | | | | | | | | | |
| geca | 0.394748 | | | | | | | | |
| learned aug | 0.761129 | 0.635337 | | | | | | | |
| recomb-1 | 0.428606 | 0.974851 | 0.620601 | | | | | | |
| recomb-2 | 0.199768 | 0.601998 | 0.317494 | 0.625078 | | | | | |
| geca + resampling | 2.27478e-25 | 6.11513e-29 | 3.30904e-24 | 1.38242e-24 | 2.19894e-27 | | | | |
| learned aug + resampling | 1.09224e-10 | 9.34816e-13 | 1.40474e-11 | 4.70624e-13 | 4.78418e-14 | 0.000137083 | | | |
| recomb-1 + resampling | 4.00039e-27 | 8.88347e-30 | 4.06546e-25 | 1.2159e-28 | 7.2465e-29 | 0.495734 | 1.35727e-05 | | |
| recomb-2 + resampling | 1.17709e-17 | 1.29429e-21 | 4.07864e-18 | 1.12332e-19 | 1.66638e-21 | 0.313143 | 0.00925477 | 0.103819 | |

Table 11: Swahili language $p$-values for paired $t$-test in PST+FUT tenses for the average $F_1$ (micro) scores over several runs without Bonferronni correction.

| | baseline | geca | learned aug | recomb-1 | recomb-2 | geca + resampling | learned aug + resampling | recomb-1 + resampling | recomb-2 + resampling |
|---|---|---|---|---|---|---|---|---|---|
| baseline | | | | | | | | | |
| geca | 0.606002 | | | | | | | | |
| learned aug | 0.000857131 | 0.00384601 | | | | | | | |
| recomb-1 | 6.27581e-05 | 0.00101351 | 0.769589 | | | | | | |
| recomb-2 | 1.75947e-05 | 0.000207507 | 0.263242 | 0.402064 | | | | | |
| geca + resampling | 2.58696e-21 | 8.85433e-19 | 4.57259e-11 | 1.33673e-11 | 1.87968e-08 | | | | |
| learned aug + resampling | 7.09377e-53 | 1.46895e-47 | 2.3846e-38 | 6.32242e-38 | 2.09274e-30 | 3.26321e-10 | | | |
| recomb-1 + resampling | 1.66361e-58 | 1.28557e-54 | 9.46035e-44 | 2.05703e-45 | 9.46848e-37 | 1.60241e-17 | 0.0749463 | | |
| recomb-2 + resampling | 2.16531e-31 | 3.52334e-25 | 7.80047e-20 | 1.08762e-19 | 1.26218e-15 | 0.0756646 | 1.52594e-06 | 2.51776e-11 | |

### G.3 GENERATED SAMPLES

All samples are randomly selected unless otherwise indicated.

### G.3.1 SCAN

In Table 12, we present three test samples from the SCAN task along with the predictions by direct inference and the conditional model trained on the augmented data with *recomb-2*. Note that the augmentation procedure was able to create novel samples whose input ($x$) happens to be in the test set (Examples 1 and 3) while $y$ may or may not be correct (Example 1).

| | Example 1 (*jump*) | Example 2 (*jump*) | Example 3 (*around right*) |
|---|---|---|---|
| Input ($x = \hat{x}$) | walk twice after jump twice | run right after jump twice | jump left and jump around right |
| True label ($y$) | JUMP JUMP WALK WALK | JUMP JUMP RTURN RUN | TURN LEFT JUMP TURN RIGHT JUMP TURN RIGHT JUMP TURN RIGHT JUMP TURN RIGHT JUMP |
| $\hat{y}$ in augmented dataset | JUMP JUMP JUMP WALK | (not generated) | TURN LEFT JUMP TURN RIGHT JUMP TURN RIGHT JUMP TURN RIGHT JUMP TURN RIGHT JUMP |
| Predicted $\hat{y}$ | | | |
| ├ direct inference | JUMP JUMP JUMP WALK | LOOK LOOK RTURN JUMP | TURN LEFT JUMP TURN LEFT JUMP TURN LEFT JUMP TURN LEFT JUMP TURN LEFT JUMP |
| └ *recomb-2* | JUMP JUMP WALK WALK | JUMP JUMP RTURN RUN | TURN LEFT JUMP TURN RIGHT JUMP TURN RIGHT JUMP TURN RIGHT JUMP TURN RIGHT JUMP |

Table 12: Comparison of generative and unconditional model predictions with and without data augmentation. The conditional model trained on augmented data (final row) is able to compensate for errors in data augmentation (Example 1) and generalize to examples never generated by the data augmentation procedure (Example 2), and often times if a sample is correctly created by the augmentation procedure, the conditional model also gets it right (Example 3).

Below are a set of samples from the learned aug (basic) model for SCAN dataset's *jump* and *around right* splits, in order:

IN: run opposite and walk opposite right twice OUT: RUN TURN RIGHT TURN RIGHT RUN TURN RIGHT TURN RIGHT WALK

IN: look around right thrice after run around thrice thrice OUT: TURN RIGHT RUN TURN RIGHT RUN TURN RIGHT RUN TURN RIGHT RUN TURN RIGHT RUN TURN RIGHT RUN TURN RIGHT RUN TURN RIGHT RUN TURN RIGHT RUN TURN RIGHT RUN TURN RIGHT RUN TURN RIGHT LOOK TURN RIGHT LOOK TURN RIGHT LOOK TURN RIGHT LOOK TURN RIGHT LOOK

IN: look opposite right twice and walk around twice OUT: TURN RIGHT TURN RIGHT LOOK TURN RIGHT TURN RIGHT LOOK TURN RIGHT WALK TURN LEFT WALK TURN LEFT WALK TURN LEFT WALK

IN: run opposite and thrice OUT: RUN TURN LEFT RUN RUN

IN: walk opposite right thrice turn turn right thrice OUT: TURN RIGHT TURN RIGHT TURN RIGHT TURN RIGHT TURN RIGHT TURN RIGHT TURN RIGHT TURN RIGHT TURN RIGHT WALK

IN: jump opposite right twice jump look around left OUT: TURN RIGHT TURN RIGHT JUMP TURN LEFT TURN LEFT JUMP TURN LEFT TURN LEFT LOOK TURN LEFT TURN LEFT LOOK

```
IN: walk around left thrice after jump left left OUT: TURN LEFT WALK TURN LEFT WALK TURN LEFT WALK
TURN LEFT WALK TURN LEFT WALK TURN LEFT WALK TURN LEFT WALK TURN LEFT WALK TURN LEFT WALK TURN LEFT
WALK TURN LEFT WALK TURN LEFT WALK TURN LEFT WALK TURN LEFT WALK TURN LEFT WALK TURN LEFT WALK TURN
LEFT WALK
IN: run opposite right twice walk run left thrice OUT: TURN RIGHT TURN RIGHT RUN TURN RIGHT TURN
RIGHT WALK TURN LEFT TURN LEFT RUN TURN LEFT TURN LEFT RUN
```

Below are a set of samples from the *recomb-1* model for SCAN dataset's *around right* split. Note that there were no samples with rare tags generated by the model for the *jump* split:

```
IN: run around right after walk around left OUT: TURN LEFT WALK TURN LEFT WALK TURN LEFT RUN TURN
LEFT RUN TURN LEFT WALK TURN LEFT WALK TURN LEFT WALK TURN LEFT WALK
IN: look around right after jump around left OUT: TURN LEFT LOOK TURN LEFT JUMP TURN LEFT LOOK TURN
LEFT LOOK TURN LEFT JUMP TURN LEFT JUMP TURN LEFT JUMP TURN LEFT JUMP
IN: look around right and jump around left OUT: TURN RIGHT LOOK TURN RIGHT LOOK TURN RIGHT LOOK TURN
LEFT JUMP TURN LEFT JUMP TURN LEFT JUMP TURN LEFT JUMP TURN LEFT JUMP
IN: walk around right and turn right twice OUT: TURN RIGHT WALK TURN RIGHT WALK TURN RIGHT WALK TURN
RIGHT WALK TURN RIGHT TURN RIGHT
```

Below are 4 samples from the *recomb-2* model for each of SCAN dataset's *jump* and *around right* splits, respectively:

```
IN: jump opposite left thrice after jump opposite left thrice OUT: TURN LEFT TURN LEFT JUMP TURN
LEFT TURN LEFT JUMP TURN LEFT TURN LEFT JUMP TURN LEFT TURN LEFT WALK TURN LEFT TURN LEFT WALK TURN
LEFT TURN LEFT WALK
IN: jump left thrice and jump left thrice OUT: TURN LEFT LOOK TURN LEFT LOOK TURN LEFT LOOK TURN LEFT
JUMP TURN LEFT JUMP TURN LEFT JUMP IN: jump opposite right and turn around left OUT: TURN RIGHT TURN
RIGHT JUMP
TURN LEFT TURN LEFT TURN LEFT TURN LEFT
IN: turn around left and jump around left OUT: TURN LEFT TURN LEFT TURN LEFT TURN LEFT TURN LEFT
JUMP TURN LEFT JUMP TURN LEFT JUMP TURN LEFT JUMP

IN: look right twice after run around right OUT: TURN RIGHT RUN TURN RIGHT RUN TURN RIGHT
RUN TURN RIGHT RUN TURN RIGHT LOOK TURN RIGHT LOOK
IN: turn right twice after look around right OUT: TURN RIGHT LOOK TURN RIGHT LOOK TURN RIGHT LOOK
TURN RIGHT LOOK TURN RIGHT TURN RIGHT
IN: look twice and run around right OUT: LOOK LOOK TURN RIGHT RUN TURN RIGHT RUN TURN RIGHT RUN TURN
RIGHT RUN
IN: walk opposite right twice and jump around right OUT: TURN RIGHT TURN RIGHT WALK TURN RIGHT TURN
RIGHT WALK TURN RIGHT JUMP TURN RIGHT JUMP TURN RIGHT JUMP TURN RIGHT JUMP
```

## G.3.2  MORPHOLOGY

Below are a set of samples from the learned aug (basic) model in SIGMORPHON format.

```
şahmiçe şahmiçende N;LOC;SG;PSS2S
karadan havaya füze karadan havaya füzel N;DAT;PL;PSS3P
ernek erneklerine N;DAT;PL;PSS3P
kiler kilerime N;DAT;SG;PSS1S
mahlep mahlebimizi N;ACC;SG;PSS1P
süzmek süzerler V;IND;3;PL;PRS;POS;DECL
âlap âlaps N;LGSPEC1;3S;SG;PRS
jöle jöleleri N;ACC;PL

envejecerse envejeciéndose V.CVB;PRS
colaxar colaxa V;IND;PRS;3;SG
pergedrer no pergedremos V;NEG;IMP;1;PL
mantear no mantees V;NEG;IMP;2;SG
flaguear no flagueen V;NEG;IMP;3;PL
malacostar malacostaría V;COND;3;SG
```

```
desinstar desinse V;POS;IMP;3;SG
concretizar no concretices V;NEG;IMP;2;SG
```

Below are a set of samples from the learned aug (basic) + resampling model.

```
şaşırmak şaşırmıyor musun? V;IND;2;PL;PST;PROG;POS;INTR
ayılmak ayılmaya V;IND;1;SG;PST;DECL
pleşmek pleşmiyor muyuz? V;IND;1;PL;PST;PROG;NEG;INTR
imciyetmek imciyetmezdeğiz V;IND;1;PL;FUT;NEG;DECL
kuvaşmak kuvaşmayacağız V;IND;1;PL;FUT;NEG;DECL
yermek yermeyeceğiz V;IND;1;PL;FUT;NEG;DECL
yarıtmak yarıtmayacağız V;IND;1;PL;FUT;NEG;DECL
kelimek kelimeyeceğiz V;IND;1;PL;FUT;NEG;DECL

trasescar trasescáis V;IND;PST;2;PL;IPFV
tronar tronar V;IND;FUT;1;SG
terzcalminar terzcalminan V;IND;PST;3;PL;IPFV
esubronizar esubronizamos V;IND;PST;1;PL;IPFV
urdir urdiremos V;IND;FUT;1;PL
conder conderemos V;IND;FUT;1;PL
florear florearían V;IND;PST;3;PL;LGSPEC1;SG
sabrordar sabrordamos V;IND;PST;1;PL;IPFV
```

Below are a set of samples from the *recomb-1* + resampling model (the best performing model in Table 2). Here we additionally annotate samples with error categories.

```
kovulmak kovulmaz mısınız V;IND;2;PL;FUT;NEG;INTR (Inflection and tags don't match.)
düşünmek düşündüler V;IND;3;PL;PST;POS;DECL (Correct and novel.)
sütmek sütmez miyiz? V;IND;2;SG;FUT;NEG;INTR (Inflection and tags don't match.)
bakmak bakmayacak mıyım? V;IND;1;PL;FUT;NEG;INTR (Inflection and tags don't match)
döndürmek döndürecek misiniz? V;IND;2;PL;FUT;POS;INTR (Correct and novel.)
türkçeleştirtmek türkçeleştirtiyor m V;IND;2;PL;PST;PROG;NEG;INTR (Wrong inflection, novel tag.)
çalmak çalmayız V;IND;2;PL;FUT;POS;DECL (Inflection and tags don't match.)
üsürmek üsürmezsin V;IND;2;SG;PST;NEG;DECL (Inflection and tags don't match.)

duplicar duplicaráis V;IND;FUT;2;PL (Correct and novel)
efundar efundan V;SBJV;FUT;3;PL (Inflection and tags don't match)
deshumanizar deshumanicas V;SBJV;PST;2;SG (Inplausible inflection.)
emular emulares V;SBJV;FUT;2;SG (Correct and also in train set.)
languidecer languidecíamos V;IND;PST;1;SG;IPFV (Inflection and tags don't match)
nominar nominamos V;SBJV;FUT;1;PL (Novel tags, incorrect inflection.)
finciar finciare V;SBJV;FUT;1;SG (Correct and novel.)
abastar abasto V;IND;PST;1;SG (Inflection and tags don't match)
```

## G.4 Attention Heatmap

Here we provide a visualization copy and attention mechanism in *recomb-2* model for SCAN experiments.

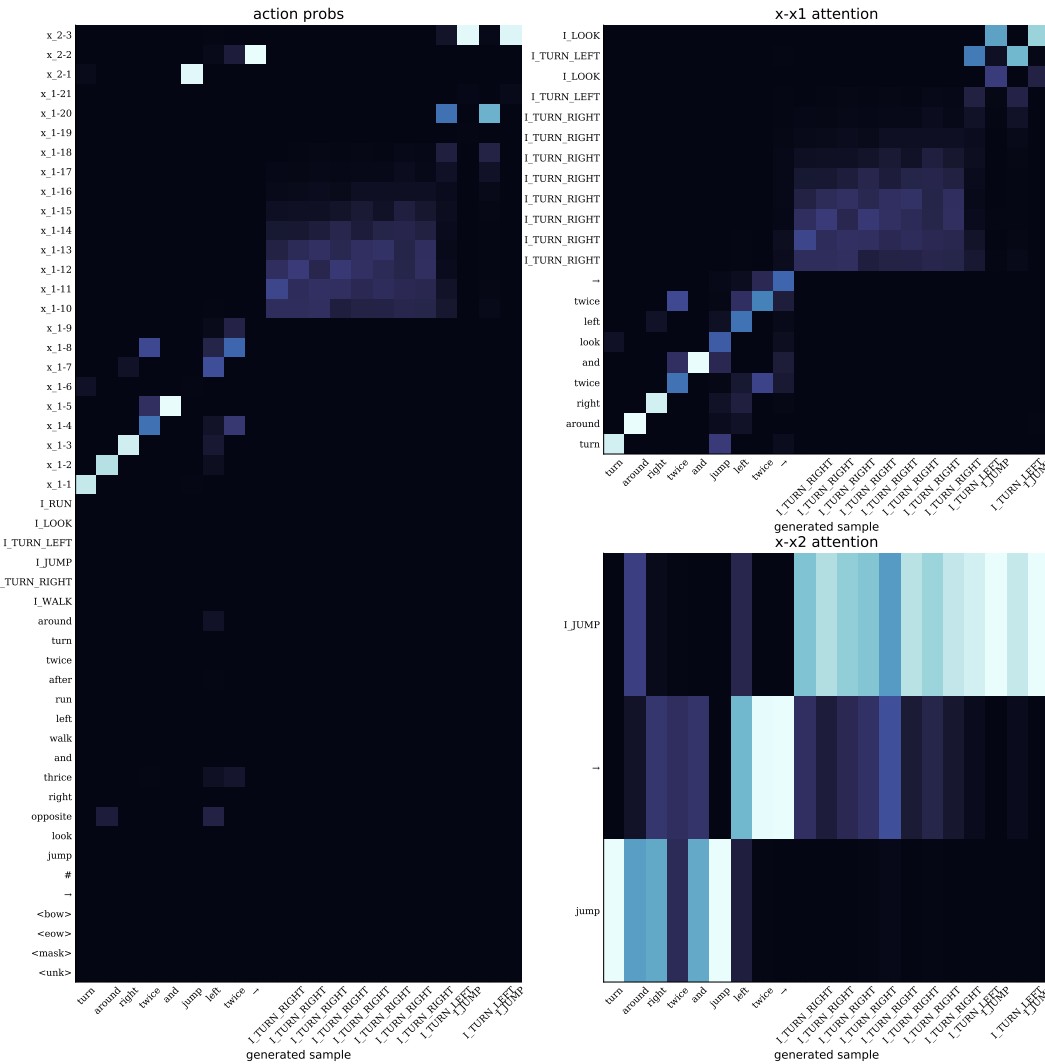

Figure 3: Generation of a sample. We plot normalized output scores on the left, and attention weights to the different prototypes on the right. The prototypes are on the y axes. The model is recomb-2 model trained on SCAN *jump* split.

## H Compute

We use a single 32GB NVIDIA V100 Volta GPU for each experiment. For every experiment, the whole pipeline which consists of training of the generative model, sampling and training of the conditional model takes less than an hour.

