# OpenReview forum: "Learning to Recombine and Resample Data For Compositional Generalization"
_ICLR.cc/2021/Conference — ICLR 2021 Poster_

### Official Review · AnonReviewer4 · 2020-10-20
**neural data augmentation for compositionality; not clear for its pros and cons compared to its non-neural counterpart.**

**Rating:** 6
**Confidence:** 3

**Review:**

####Summary:
To tackle situations where compositionality is mostly required at inference time, the paper proposes a novel data augmentation method with an RNN based generator (recombination); to make the generator generate highly compositional patterns, the paper proposes a resampling method. The methods have been tested on two benchmarks focusing on the issue, SCAN and morphological analysis. The system performs on par with recently proposed GECA for SCAN and favorably to GECA on morphological analysis.

The two datasets have some “toy flavor”, while SCAN great favors example combination (with recomb-2 performs much better than recomb-1), recomb-1 seem to perform better for morphological analysis dataset, leaving questions about how to choose the exact models in general.

####Pros:
The paper proposes the first RNN based neural generator to perform data augmentation for “extreme” compositionality inference. The paper has explained and empirically showed that this learned generator needs a resampler. With these two elements, the approach performs on par with recently proposed GECA (where the data is not augmented via a neural generator) on two datasets.

####Question:
1. Comparison with GECA: I can read from the paper that the performance is on par with GECA. However, I am unable to grasp nuances, leaving important questions untouched such as: In what scenarios do we expect the model to perform better than GECA? In experimental details, the slightly better performance in Table 2, can it be attributed to finer generation powered by the RNN generator? Why does Recomb-2 perform less well than GECA in SCAN?

2. A uniform framework for resampling
Different recombinations perform more or less favorably across different datasets. While the exact choice depends on the dataset characteristics, a framework will be more attractive if it can perform well on different scenarios. Could the authors list some possible approaches to automatically choose this hyper-parameter please?

####Minor Comments:
The paper mentions in several places symbolic scaffolding without citations, literature is certainly rich here, e.g. [1,2] are papers integrating symbolic constraints for semantic parsing. There are also neural architectures that particularly target to ensure some symbolic famous properties such as [3].

The authors say in the introduction that the approach (Andreas, 2020) is task specific which seems not correct. In fact, it can be applied to a large range of NLP problems (e.g. all the experiments in this paper compares to the approach GECA))

In section 3, it says that “the use of a continuous latent variable appears to make no difference”, I would suggest to precise “make no difference” as “make no difference in prediction performance“ as the latent variable can facilitate some generation control shown in (Guu et al. 2018).

In this paper, GECA is first introduced in section 5. I would recommend to put the citation around it (Andreas, 2000) although previously cited.

[1] Sequence-based structured prediction for semantic parsing, Xiao et al. 2016
[2] A syntactic neural model for general-purpose code generation, Yin and Neubig 2017
[3] Making Neural Programming Architectures Generalize via Recursion, Cai et al. 2017

####Authors have engaged in the discussion, clarified questions about the paper and addressed comments in its newest revision. I have consequently revised my score from 5 to 6.

---

> ### Author Response · Authors · 2020-11-16
> **Thank you very much for the thorough review!**
>
> Answers to your questions:
>
> **1) How to choose the models in general? In what scenarios do we expect the model to perform better than GECA?  Can we attribute slightly better performance in Table 2 to finer generation powered by the RNN generator?**
>
> The improvements due to GECA and learned augmentation are at least somewhat orthogonal (see response to R3), so it’s not the case that one approach strictly dominates the other, and we don’t have to choose just one!
>
> Slightly more formally, GECA can be viewed as computing the closure of a training set under the assumption that the training set is *substitutable* (Clark and Eyraud, JMLR 2007). When this assumption is violated, GECA will introduce systematic errors into the augmented training set. As a concrete example, the formal language a^nb^n is not substitutable (see p1738 of C&E), and applying GECA to a dataset of strings a^nb^n will also produce strings a^nb^m for n != m. However, RNNs like our "learned aug (basic)" model are capable of correctly modeling a^n b^n and generalizing to new n (see Weiss et al. ACL 2018 or Gers & Schmidhuber 2001). So we have a concrete example of a language where learned augmentation is more expressive than GECA.
>
> **2) Why does recomb-2 perform less well than GECA in SCAN?**
>
> Recomb-2 performs less well only in the around right split of SCAN. The substitution rule that GECA infers is able to precisely change every "X left" with "X right"  and corresponding "I_LEFT" with "I_RIGHT" in the output, so is particularly well-suited for this split. There is somewhat more noise in the learned model’s flexible generation procedure; this is necessary to accommodate more natural data like Sigmorphon.
>
> **3) Task specificness of GECA?**
>
> Thanks for the suggestion. We’ll adjust this language in the final version of the paper; our main point here is that GECA makes specific generative assumptions about sequence data that may be violated by real-world datasets. Another advantage of learned augmentation (though not considered in this paper) is the possibility of applying learned recombination to non-sequential data (e.g image and caption pairs).

---

> > ### Comment · AnonReviewer4 · 2020-11-16
> > **Thank you for the feedback**
> >
> > The authors have successfully addressed my questions on the paper by providing two new elements (listed in authors' general comments). I would really like to see those elements in the final version of the paper (and I think other readers will appreciate it as well) together with addressing other comments by reviewers.
> >
> > Once those have been done (not yet at this stage), I will be glad to raise by score to 6.

---

### Official Review · AnonReviewer1 · 2020-10-28
**Shows unstructured models can generalize on compositional data via a learned data augmentation procedure**

**Rating:** 7
**Confidence:** 3

**Review:**

Summary:
* Motivated by the fact that certain datasets require modeling compositional phenomena, the lack of flexibility of highly structured models, and the strong performance of large unstructured models on unstructured data, this paper approaches the problem of getting unstructured models to generalize on compositional data.
* Prior work showed that a simple rule-based data augmentation approach could allow unstructured models to generalize on compositional data. This paper demonstrates that a learned data augmentation strategy can be as effective at encouraging generalization as a rule-based one.

Contributions:
* Extends the prototype+edit model (Guu et al 2018) to a recombinator model with multi-source copy attention.
* Proposes a resampling scheme for upweighting rare examples.
* Obtains results comparable to strong rule-based data augmentation baseline GECA on two datasets, demonstrating that the combination of both resampling and recombination is effective.

Strengths:
* Clearly written. The method is simple and is broken down cleanly into recombination and resampling.
* Ablation studies support the need for both recombination and resampling.
* Sufficient performance to support the claim. The approach manages to match the performance of GECA with a learned method, while remaining more flexible.

Weaknesses:
* Neighbourhood heuristics are one of the last remaining applications of manual rules in the method, but seem necessary for computationally feasible training.

Decision: Accept
* Problem is important: Whether unstructured models can generalize on structured data has implications for whether or not to move towards more structured models. This paper provides experimental evidence that unstructured models can generalize on structured data with a data augmentation procedure that uses fewer manually specified rules than previously shown. This provides a path forward by continuing to iterate on the augmentation procedure.

Questions:
* Given that GECA is feasible on both datasets, would there be benefit to combining the examples from the learned augmentation strategy with those from GECA? In other words, do the different augmentation strategies result in orthogonal improvements?
* Why does resampling hurt GECA on SCAN?

Suggestions:
* Both datasets are quite small. The story could be strengthened by demonstrating the method scales better than GECA by applying it to a larger dataset as well, such as a translation dataset.

Nit:
* w is overloaded to both be a value of d (eqns 12, 13, 17) as well as the weighting function (eqns 1, 18)

---

> ### Author Response · Authors · 2020-11-16
> **Thanks for the positive review and detailed feedback!**
>
> Here are answers to questions raised by the reviewer:
>
> **1) Combining the examples from the learned augmentation strategy with those from GECA? Are the improvements orthogonal?**
>
> Thanks for the great suggestion! After the review, we tried ensembling GECA with each neural model (0,1,2 proto) in Sigmorphon. The best Turkish result is changed from 77% to 78%, the best Spanish results is changed from 72% to 74%, Swahili is unchanged. These results suggest that gains from GECA and learned augmentation are at least partly orthogonal. We will include these results and discuss them in the final version.
>
> **2) Why does resampling hurt GECA on SCAN?**
>
> We believe this is due to resampling a larger fraction of examples that are malformed as well as the greater correlation between x and the noise applied to y. Also note that the std values for these experiments are high.
>
> **3) Larger Dataset?**
>
> We agree that this is an important direction for future work! See response to R2 for more detail.

---

### Official Review · AnonReviewer2 · 2020-10-28
**Interesting work with nice findings. Would be valuable to know the applicability to real-world data**

**Rating:** 7
**Confidence:** 4

**Review:**


Summary:


The paper proposes an interesting approach to systematically generate new examples and augment the training data with these examples. The goal is to target rare and unseen sequences of text or instructions with this augmentation.
In particular, it proposes learning to copy parts of the reference examples. The approach is based on prototype-based models where every training example can be explained by at least one other example and a parametric rewriting operation. They show that these models do not perform well when facing complex (and rare) composition events and propose a recombination addition to address this issue.

I like the idea of systematically augmenting training data targeting rare and unseen subsequences. I give an "accept" to this work because of its novelty and contribution (see pros below).
My minor concern is about the impact and/or usefulness of this work when dealing with real-world datasets and more complicated tasks as well as some clarity issues (see cons below). Hopefully, the authors can address my concern in the rebuttal period.

 ##########################################################################

Pros:

1. The paper addresses one of the interesting and important shortcomings of current neural models: the ability to generalize to rare and unseen sequences. I find this problem important to investigate and applicable to many areas of research.

2. The proposed approach is flexible and practical to use. The design of the prototype-based data augmentation method is reasonable and interesting.

3. This paper provides comprehensive experiments, including both qualitative analysis and quantitative results, to show the effectiveness of the proposed framework.


##########################################################################

Cons:


1. In Table 2, we see F1 score for morphological analysis. It is not entirely clear to me why the results in the NOVEL section of the table is very close to ALL section. NOVEL shows model accuracy on examples whose exact tag set never appeared in the training data and I expected a bigger gap in the performance.

2. There is a big gap between the performance of 1- and 2-prototype models. Do the authors know what explains this gap? Have the authors explored the higher order of prototype recombinations?

3. What are the challenges of applying this approach to real-world data sets for instance in machine translation? I will suggest the authors discuss the implications and possible shortcomings of such an approach when dealing with natural (and potentially long) sequences of text. The definition of unseen subsequences and compositional learning will be more complex there.

4. It is not clear to me what the authors mean by "hints". Is the complete sequence counts as a hint? Or only the subsequence that was identified as rare?


#########################################################################

Some typos:


(1) Relevant literature that was not mentioned in this paper:
         https://arxiv.org/abs/1705.00440
         https://arxiv.org/abs/1801.02929

(2) Equation (31): tex formatting issue.

(3) Typo in page 15 section G1: Fig. Table 2 shows -> Table 2 shows

---

> ### Author Response · Authors · 2020-11-16
> **We appreciate the positive review and detailed suggestions for improving the paper!**
>
> Here are answers to questions raised by the reviewer:
>
> **1) What does ‘hints’ mean?**
>
> By "hints", we mean the number of (x, y) pairs exhibiting the novel tense; "hints=k" means that only k training examples have the PAST tense and only k data points have the FUTURE tense.
>
> **2) Why are the results in the NOVEL section of the table very close to the ALL section?**
>
> With hints=8, most of the samples in the FUT+PST test set feature novel combinations of morphological tag combinations, whereas few of the OTHER examples (and none in Spanish) feature novel tags. So we expect close results between ALL and NOVEL in the FUT+PST (because the two evaluation sets largely overlap), and we expect high variance on OTHER in NOVEL split since it contains few examples.
>
> **3) Gap between 1-proto and 2-proto; and higher-order recombinations (n>2)?**
>
> We observe a big gap in SCAN because 1-proto and standard language models cannot generate compositional examples that the task requires, but 2-proto can. We agree that the n > 2  case is an important challenge for future work; here we think a promising direction is to let neighborhoods be fully learned.
>
> **4) Real world datasets such as MT?**
>
> We agree! We emphasize that the morphology dataset is more realistic than many existing synthetic datasets used for studying compositional generalization. However, extending this work further to sequence-to-sequence problems with hundreds of thousands or millions of examples is another important direction for future work, and could be done by e.g. combining this approach with an existing retrieval based language model like REALM (Guu et al. 2020).
>
> We tried to fix the typos in the new version of the paper posted here; we’ll post a final version with an expanded discussion of related work once we have finished writing it.

---

### Official Review · AnonReviewer3 · 2020-10-29
**Clearly written and motivated contribution to data augmentation with solid empirical results**

**Rating:** 8
**Confidence:** 4

**Review:**

This paper presents a prototype-based method for data augmentation based on a generative model without rule/template based requirements. The generative model creates new input-output pairs from training fragments (recombination: rewrite model conditioned on multiple examples), and samples in low-density places (rare words) of the training data (resampling). Empirical results show that the in combination recombination and resampling perform on par with a recently introduced rule-based method, GECA.
Experiments are conducted on two compositional generalization tasks: SCAN and sigmorphon.

The paper is very clearly written and motivated, doing a good job in presenting the recent and past pertinent literature. The problem addressed is of great interest, and the two proposed contributions of resampling and recombination are likely to be useful to further research in data augmentation.

The extension of prototype models to multiple examples is a promising step, but depending on the task leaves open questions.
The empirical results on SCAN are strong. The results on Sigmorphon are strong in the sense of obtaining comparable accuracy to simple rule-based approach, which itself is very simple and has many incorrect examples it constructs, but do not clearly outperform it.
Granting resampling is a contribution, wouldn’t the proper comparison be GECA resampling with the recomb-1 or -2, since those include resampling as well? In that case the Sigmorphon performance is much closer to each other for the two methods.

Taking the performance jump from 1 to 2 prototypes on SCAN as potential for further jumps, why restrict to n=2? And related, why do you think -2 outperforms -1 in some instances on Sigmorphon?

---

> ### Author Response · Authors · 2020-11-16
> **Thank you for the positive review!**
>
> Answers to the reviewer's questions:
>
> **1) Comparison of GECA and Recomb models**
>
> We definitely agree that the most informative comparison in Table 2 is between GECA+resampling and recomb+resampling---this is why we focused our evaluation (e.g. for the "novel" condition at the bottom of T2) on these two models. We respectfully disagree with the claim that our approach obtains "comparable accuracy to simple rule-based approach": recomb+resampling significantly outperforms GECA+resampling in multiple experiments (All/Swahili, Novel/Swahili and Novel/Turkish) while matching it in the rest: recombination is measurably better even compared to the improved version of GECA presented in this paper.
>
> **2) Why restrict to n=2?**
>
> The n > 2 case would certainly be a natural extension of this approach; one of the main research questions here is how to construct a natural neighborhood function of the kind described for the n = 2 case in this paper. Allowing these neighborhood functions to be fully learned is an important challenge for future work.
>
> **3) Why does n=1 outperform  n=2 in some instances on Sigmorphon? (The review asks the opposite question, but we assume it’s a typo.)**
>
> Our intuition is that learning to recombine is a more complex task than learning to edit. When the model has enough data to robustly learn this skill, it can generate complex novel samples as seen in SCAN. The morphology datasets provide fewer samples for learning complex recombination strategies, and modifying individual examples turns out to be quite helpful on its own.

---

### Author Response · Authors · 2020-11-16
**Comment for Initial Reviews**

We thank all the reviewers for their detailed feedback on the submission! We are glad they found the paper’s description of “the first RNN based neural generator to perform data augmentation for ‘extreme’ compositionality” [R4] to be clearly written [R1, R3], of scientific interest [R1, R2, R3], and convincingly evaluated on the domains studied [R2, R1, R4]. We’ve provided replies to individual reviewer comments; some features of general interest are:
- a new set of experiments showing that GECA and learned recombination can be combined to give even better performance in Spanish and Turkish [response to R1]
- a formal example (due to Clark & Eyraud 07) of a language that cannot be modeled with GECA but can be modeled with learned augmentation [response to R4]

We’ve uploaded a revised addressing most of the edits suggested by reviewers; we are currently working on expanding the related work section and will upload another revision when that’s done. Thanks again!

---

### Decision · Program_Chairs · 2021-01-07
**Final Decision**

**Decision:**

Accept (Poster)

**Comment:**

The paper addresses generalization to compositions of rare and unseen sequences. It proposes an unstructured data augmentation, that achieves comparable generalization to structured approaches (e.g. using grammars). The idea is based on recombining prototypes and oversampling  in the tail.

The paper provides a novel approach to an important problem. All four reviewers recommended accept.